# Expanding the Phenotypic and Genotypic Spectrum of Bietti Crystalline Dystrophy

**DOI:** 10.3390/genes12050713

**Published:** 2021-05-10

**Authors:** Mariana Matioli da Palma, Fabiana Louise Motta, Mariana Vallim Salles, Caio Henrique Marques Texeira, André V. Gomes, Ricardo Casaroli-Marano, Juliana Maria Ferraz Sallum

**Affiliations:** 1Department of Ophthalmology, Federal University of São Paulo—UNIFESP, São Paulo, SP 04023-062, Brazil; marimatioli@yahoo.com.br (M.M.d.P.); fabiana.louise@gmail.com (F.L.M.); marivallim@yahoo.com.br (M.V.S.); caiomtex@gmail.com (C.H.M.T.); rcasaroli@ub.edu (R.C.-M.); 2Instituto de Genética Ocular, São Paulo, SP 04552-050, Brazil; 3Instituto Suel Abujamra, São Paulo, SP 01525-001, Brazil; andremv@uol.com.br; 4Department of Surgery & Hospital Cínic de Barcelona, School of Medicine, Universitat de Barcelona, 08007 Barcelona, Spain

**Keywords:** bietti crystalline dystrophy, CYP4V2 protein, genetic testing, missense mutation, insertion-deletion mutation

## Abstract

The rare form of retinal dystrophy, Bietti crystalline dystrophy, is associated with variations in *CYP4V2*, a member of the cytochrome P450 family. This study reports patients affected by typical and atypical Bietti crystalline dystrophy, expanding the spectrum of this disease. This is an observational case series of patients with a clinical and molecular diagnosis of Bietti crystalline dystrophy that underwent multimodal imaging. Four unrelated patients are described with two known variants, c.802-8_810del17insGC and c.518T > G (p.Leu173Trp), and one novel missense variant, c.1169G > T (p.Arg390Leu). The patient with the novel homozygous variant had the most severe phenotype resulting in macular hole formation and retinal detachment in both eyes. To the best of our knowledge, there is no association of these features with Bietti crystalline dystrophy. Patient 1 was the youngest patient and had the mildest phenotype with crystals in the retina without chorioretinal atrophy and visual complaints. Patients 2 and 3 presented with fewer crystals and chorioretinal atrophy. These three patients presented a classic phenotype. The fourth patient presented with an atypical and severe phenotype. This study reveals a new genotype and new phenotype associated with this disorder.

## 1. Introduction

Bietti crystalline dystrophy (BCD) (OMIM210370) is an inherited recessive disorder characterized by crystalline deposits in the retina and sometimes at the corneoscleral limbus [1,2]. Pathogenic variants in the *CYP4V2* gene (OMIM#608614) were identified as disease-causing [3]. The protein encoded by this gene is a member of the cytochrome P450 family. It has been suggested that CYP4V2 proteins are implicated in the lipid recycling system between the retinal pigment epithelium (RPE) and outer photoreceptor segments, which is essential for maintaining visual acuity [1,4]. This gene is expressed in the human heart, brain, lung, liver, kidney, placenta, retina, and lymphocytes [1,3,4]. Histopathology showed lipid inclusions in lymphocytes, skin fibroblasts; however, clinically significant abnormalities remain only in the eye [3,5,6].

BCD causes nyctalopia, decreased visual acuity, and visual field constriction, similar to other forms of retinal degeneration [1,2,3,4]. The crystalline deposits seen early in the retina are the hallmark of BCD. However, these crystalline deposits can also appear in other retinal diseases such as reticular pseudodrusen, retinitis *punctata albescens*, cystinosis, and Sjögren-Larsson syndrome [7]. Perimacular yellow-white dots are also seen in Alport syndrome [8]. With disease progression, the yellow-white crystals disappear [7]. The advanced stage [9] of BCD is characterized by extensive chorioretinal atrophy that can be similar to other inherited retinal diseases such as choroideremia, retinitis pigmentosa, fundus *albipunctatus*, and gyrate atrophy [1]. The overlap of clinical phenotype, in both early and late stages of the disease, emphasizes the importance of molecular testing in establishing a precise diagnosis and has an impact on disease management [1].

The current study describes four patients with BCD from the Federal University of São Paulo and *Instituto de Genética Ocular* in Brazil. One of them with an atypical phenotype associated with a novel homozygous missense variant c.1169G > T (p.Arg390Leu) in the *CYP4V2* gene.

## 2. Materials and Methods

This study was performed in accordance with the Declaration of Helsinki and protection of the patient’s identity and was approved by the Research Ethics Committee of the Federal University of São Paulo (number 1191.0071.10). This is a case series of patients with a molecular diagnosis of BCD. The medical records were reviewed. The patients underwent detailed ophthalmic exams including best-corrected visual acuity (BCVA), slit-lamp exam, and multimodal retinal imaging: fundus images (VISUCAM 500, Zeiss, Oberkochen, Germany), FAF (HRA2, Heidelberg Engineering, Heidelberg, Germany), OCT (Spectralis, Heidelberg Engineering, Heidelberg, Germany). Electroretinogram was performed in patient 1 in accordance with the International Society for Clinical Electrophysiology of Vision (ISCEV) [10]. The next-generation sequencing panel, targeting inherited retinal diseases, including genes that cause chorioretinal dystrophies such as *CHM*, *OAT, RPGR, CYP4V2,* and more than 200 genes, was performed to establish a genetic diagnosis in patients 1, 3, and 4. *CYP4V2* gene sequencing was performed by using Sanger sequencing in patient 2.

Nucleotide and protein changes were described as recommended by the Human Genome Variation Society (HGVS) and based on NM_207352.4 and NP_997235.3 reference sequences. All variants found were compared with variants listed in the Human Gene Mutation Database (HGMD) [11] and ClinVar [12]. VarSome Software (Saphetor, Lausanne, Switzerland) was also used [13].

## 3. Results

The findings of the four patients are summarized in Table 1.

### 3.1. Case 1

A 19-year-old woman without complaints was referred because of fundus findings observed at age 12 years. Her BCVA was 20/20 in both eyes (OU). The patient was the only affected member of her family; her maternal grandparents were from China, and her paternal grandparents were Japanese. The slit-lamp exam was unremarkable. Fundus exam showed crystalline deposits in the retina; optic disc and retinal vessels were normal. Fundus autofluorescence showed hypoautofluorescent dots throughout the posterior pole. Spectral-domain optical coherence tomography (OCT) showed spherical intraretinal hyperreflective lesions, which confirmed the presence of intraretinal crystals (Figure 1A–C). The full-field electroretinogram was normal at age 12. Molecular testing identified a homozygous indel variant c.802-8_810delinsGC found more frequently in Asian patients [7].

### 3.2. Case 2

A female patient, 54 years old, with a Japanese background and a history of decreased visual acuity since she was 12 years old. Her BCVA was 20/50 in the right eye and 20/400 in the left eye. The slit-lamp exam showed intraocular lenses OU. Fundus exam showed normal optic disc, normal retinal vessels, crystalline deposits in the central retina, diffuse chorioretinal atrophy, and pigment clumps in both eyes. Fluorescein angiography showed a central hyperfluorescent island surrounded by multiple patchy coalescent hypofluorescent areas located throughout the degenerative lesions. OCT showed extensive outer retinal atrophy with few intraretinal crystals and outer retinal tubulations in both eyes (Figure 1D–F). Molecular testing found a missense variant c.518T > G (p.Leu173Trp) described by Lin et al. [14] and a deletion c.802-8_806del described by Li et al. [3].

### 3.3. Case 3

A 69-year-old female patient diagnosed first with retinitis pigmentosa was referred for genetic testing. She was referred to parental consanguinity. Her BCVA was 20/150 in both eyes. The slit-lamp exam showed intraocular lenses OU. Fundus exam showed normal optic disc, cup/disc 0.4, mild arterioles attenuation, crystalline deposits in the central retina, diffuse chorioretinal atrophy, and pigment clumps in both eyes. Fundus autofluorescence showed nummular areas of atrophy (Figure 1G,H). Molecular testing identified a known homozygous missense variant c.518T > G (p.Leu173Trp), the same variant found in patient 2.

### 3.4. Case 4

A Portuguese 59-year-old man who moved to Brazil at 9 years of age was diagnosed with retinitis pigmentosa when he was 34 years old. He denied family history. He underwent posterior vitrectomy and cataract surgery due to a retinal detachment in the right eye at the age of 40 years. He was complaining of progressively decreasing vision during the previous five years. His BCVA was light perception in the right eye and hand movements in the left eye. The slit-lamp exam showed an intraocular lens in the right eye and a sclerotic nuclear cataract in the left eye without crystalline deposits at the corneal limbus. Fundus exam showed extensive areas of retinal and choroidal atrophy in both eyes, silicone oil in the right eye, and a macular hole with retinal detachment in the posterior pole of the left eye (Figure 2A,B). OCT showed extensive atrophy and diffused thinning of all retinal and choroidal layers OU (Figure 2C,D) and a retinal detachment in the left eye (Figure 2E). Because he had no fixation, OCT was not focused on the macula in the left eye, and exams such as microperimetry and fundus autofluorescence could not be performed. The clinical features were inconclusive for establishing a correct diagnosis. Crystals were absent from his exam. The differential diagnosis was retinitis pigmentosa, choroideremia, and BCD. Molecular testing identified only a homozygous variant c.1169G > T (p.Arg390Leu) in CYP4V2 that had not been described previously.

## 4. Discussion

Bietti crystalline dystrophy mainly affects Asians [1,4]. The Brazilian population is ethnically very heterogeneous [15]. The largest Japanese population outside of Japan lives in Brazil, mainly in São Paulo state [16]. The estimated prevalence of rare BCD is up to 1 in 67,000 individuals [4]. The pathogenic variant in patient 1 (c.802-8_810delinsGC) has been reported previously [17]. This is the most common pathogenic variant in East Asian patients due to a founder effect [7,18,19,20]. This pathogenic variant is the result of a combination of the insertion of two nucleotides and deletion of 62 amino acid-encoding exon 7 [18]. The absence of exon 7 eliminates the essential parts of the protein structure and disrupts enzyme activity [1,3]. A more severe phenotype based on electrophysiological testing when patients had splice site mutation causing skipped exons on both alleles compared to patients without biallelic splice was reported [18]. Further, a statistically significant early age of onset between the patients with compound heterozygous c.802-8_810delinsGC mutation compared to those without c.802-8_810delinsGC was also documented [19]. However, the majority of authors did not establish any clear genotype-phenotype correlation [7,21,22].

Patients 2 and 3 presented with the same missense variant c.518T > G (p.Leu173Trp) located at exon 4. This variant was described in the literature for the first time in a 54-year-old homozygous Japanese female patient presenting suitable visual acuity in a homozygote state [14]. The Leu173 amino acid is highly conserved among eukaryotic homologs. The leucine (Leu) change at position 173 to tryptophan (Trp) may result in a structural change affecting the proper folding of the CYP4V2 protein [14]. Patient 2 and patient 3 presented a classic phenotype of tiny yellow crystals and chorioretinal atrophy in the fundus exam. Both patients also presented with pigmented spicules in both eyes. According to chart review, there were no crystals deposits at corneal limbus different from the patient previously described [14]. However, as limbal corneal crystals deposits are very subtle, they can be unnoticed even by experienced ophthalmologists [1]. They can also be absent because CYP4V2 protein is less expressed in the cornea than in the retina [19,23].

The missense variant c.1169G > T (p.Arg390Leu) found in patient 4 has never been reported. It is absent in all populations of the gnomAD database and was classified as disease-causing by nine pathogenicity predictors (DANN, GERP, FATHMM, LRT, MutationAssessor, Mutation Taster, PROVEANS, FATHMM-MKL, and SIFT) [24,25,26]. Furthermore, the Arg390 amino acid is highly conserved, and other missense variants have already been reported at the same codon (p.Arg390His and p.Arg390Cys) [17,27]. In addition, more than 35 pathogenic and likely pathogenic missense variants have been identified throughout all 11 exons of *CYP4V2* [11]. More than 80% of the mutant alleles are located in exons 7, 8, and 9 [27]. Based on ACMG [13] criteria and aforementioned information, this novel variant in exon 9 of the *CYP4V2* gene was classified as likely pathogenic.

To the best of our knowledge, there is no report of macular hole or retinal detachment in patients with BCD, as shown in the OCT of our patient 4. A giant macular hole has already been described in another coalescing fleck retinopathy, Alport syndrome [8]. The OCT findings evaluate the severity of BCD progression. In the early stage, there is a loss of the interdigitation zone and disruption in the ellipsoid zone. Outer retinal atrophy and RPE loss occurs as the severity increases. It is known that there is a relationship between age and the progressive loss of the chorioretinal layers and function [4]. The younger patient had a milder phenotype, as expected. It is also expected that the disease progresses to posterior chorioretinal atrophy, diminution of the yellow-white crystals, and slowly deteriorating vision leading to legal blindness in the fifth or sixth decades of life [1]. However, there is a well-documented high phenotypic variability [1,21]. Factors such as diet [1,11,13] may affect this variable phenotype because mutations in the *CYP4V2* gene affect lipid metabolism [3,17,21,28]. The patient with novel missense mutation presented with the worst vision reinforcing the pathogenicity of the p.Arg390Leu. It was not the older patient that had the most severe phenotype in our cohort.

There are limitations in the present study. The retrospective analysis of the Federal University of São Paulo and Instituto de Genética Ocular database from 1998 until 2020 found only four Brazilian patients with variants in *CYP4V2* of a total of 2299 individuals with inherited retinal dystrophies. The fourth patient underwent a retinal dystrophy panel that included genes that cause other crystalline retinopathies such as retinitis *punctata albescens (RLBP1)*, and fundus *albipunctatus (RDH5)*, and genes that cause other chorioretinal diseases such as choroideremia (*CHM*), gyrate atrophy (*OAT*), and retinitis pigmentosa (*RPGR*). *CTNS*, *ALDH3A2*, *COL4A5* that can cause cystinosis, Sjogren-Larson syndrome, and Alport syndrome were not tested. Only the novel variant c.1169G > T was identified. Additional cases with the same variant are missing to compare with our findings. Functional assessment of the novel missense variant may further elucidate its pathogenicity. Segregation analysis is not available in our cohort of patients. However, our results expand the phenotypic and genotypic spectrum of this ultra-rare disease.

## Figures and Tables

**Figure 1 genes-12-00713-f001:**
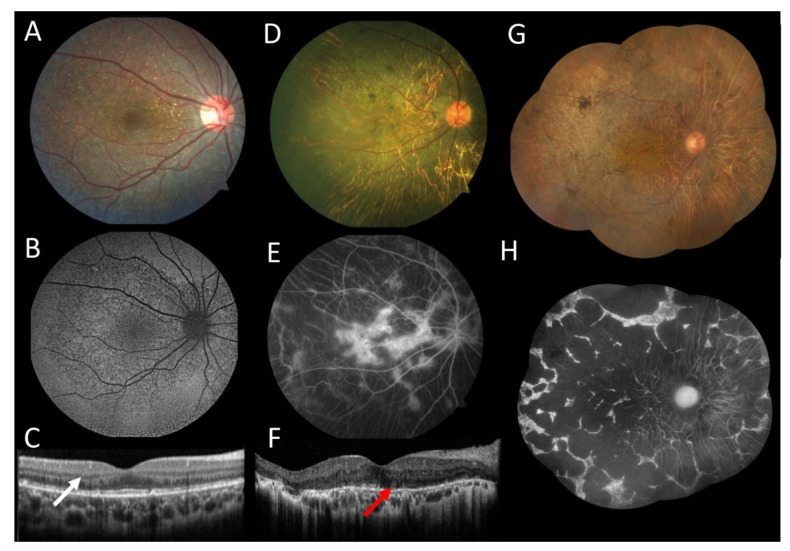
Multimodal imaging of female patients with Bietii crystalline dystrophy. (**A**–**C**) Patient 1 at age 19: (**A**) Color fundus photograph of the right eye showed crystalline deposits throughout the central retina. (**B**) Autofluorescence showed hypoautofluorescent dots representing the areas of atrophy. (**C**) The horizontal line scan from the optical coherence tomography (OCT) in the right eye showed spherical intraretinal hyperreflective crystals (white arrow). (**D**–**F**) Patient 2 at age 54: (**D**) photography of the right eye exhibited chorioretinal atrophy and pigment clumps with few crystalline deposits. (**E**) Fluorescein angiography showed multiple patchy coalescent hypofluorescent areas located throughout the degenerative lesions. (**F**) The horizontal line scan from the OCT in the right eye showed extensive outer retinal atrophy with few intraretinal crystals and outer retinal tubulations (red arrow). (**G**,**H**) Patient 3 at age 69: (**G**) color fundus montage of the right eye showed crystalline deposits, diffuse chorioretinal atrophy, and pigment clumps. (**H**) Fundus autofluorescence exhibited nummular anthropic centered.

**Figure 2 genes-12-00713-f002:**
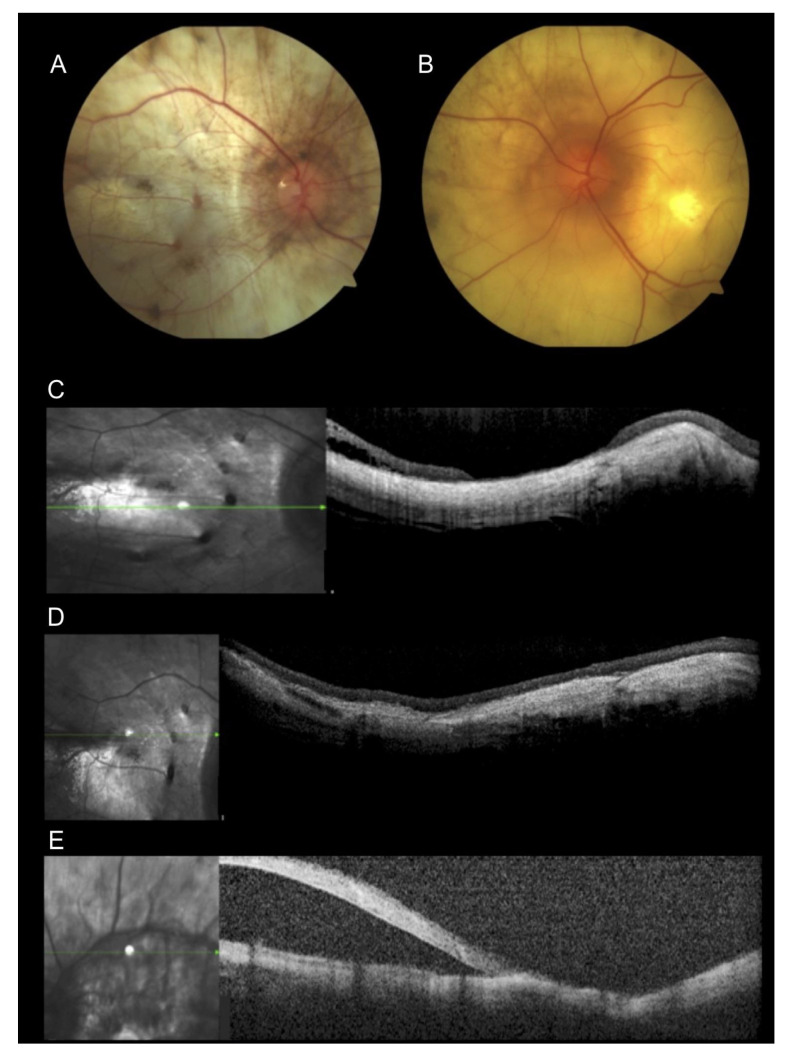
Multimodal imaging of a 59-year-old patient. (**A**,**B**) Fundus image presented extensive chorioretinal atrophy. (**A**) Fundus photography of the right eye with silicone oil and retinal attached at the central retina. (**B**) Fundus photography of the left eye showing macular hole and retinal detachment in the posterior pole. (**C**,**D**) Optical coherence tomography (OCT) scans of the right eye showed retinal and choroidal atrophy. (**E**) Horizontal OCT scan showed atrophy and retinal detachment in the left eye.

**Table 1 genes-12-00713-t001:** Molecular and clinical features of patients with Bietti crystalline dystrophy.

Patient	Age, Sex	c.DNA Change in *CYP4V2*	Protein Change	BCVA OD OS	Crystalline Deposits	OCT Findings
1	19y, female	c.802-8_810delinsGC homozygous	p.?	20/20 20/20	Retina	Intraretinal hyperreflective crystals
2	54y, female	c.518 T > G c.802-8_806del	p.Leu173Trp p.?	20/50 20/400	Retina	Outer retinal atrophy, few intraretinal crystals, and tubulations
3	69y, female	c.518T > G homozygous	p.Leu174Trp	20/150 20/150	Retina	Not available
4	59y, male	c.1169G > T homozygous	p.Arg390Leu	LP HM	None	Extensive atrophy and diffuse thinning. Central retinal detachment

BCVA: best-corrected visual acuity LP: light perception HM: hand motion.

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
