# Peer review of "Expanding the Phenotypic and Genotypic Spectrum of Bietti Crystalline Dystrophy"

_genes, 2021, doi:10.3390/genes12050713_

Round 1
Reviewer 1 Report
I think this is a well-written and fairly meaningful report. But, there are a few questions and considerations. Details are described below.
Line 61,
Did you perform whole exome sequencing or targeted exome panel? The Illumina Nextera Rapid Capture Exome Enrichment kit covers the entire exome area. The commercial next-generation sequencing panel appears to mean a gene panel targeting inherited retinal diseases (IRD). I think you need a more detailed description. For example, it is better to describe whether the panel and analysis pipeline includes IRD genes such as OAT, CHM and COL2A1(Stickler syndrome with posterior chorioretinal atrophy).
line 128-129,
Genetic testing also ~.
It would be better to remove this awkward sentence that does not match the overall content. Instead, the authors can describe the presence or absence of family history, systemic diseases, and the results of family genotyping (some homozygous mutations on NGS may be due to exon or total gene deletion in one allele).
Line 181,
Even if the variant is predicted to be pathogenic or likely pathogenic in ACMG, this is just a prediction. Therefore, it cannot be determined that the new variant is the cause of the disease only by this prediction. In general, researchers report new phenotypes in patients with well-known pathogenic variants, or new pathogenic variants in patients with typical disease phenotypes. Authors should be cautious about describing a new variant found in a single patient as a cause of new phenotypes. For example, in Asians, even severe high myopia may be accompanied by diffuse chorioretinal atrophy and RD similar to case 4. It is necessary to describe the differential diagnosis, how the authors exclude other chorioretinal diseases, and what genes were excluded by genetic analysis. It would be better to add this point to the study limitations.
Author Response
Reviewer 1
Line 61,
Did you perform whole exome sequencing or targeted exome panel? The Illumina Nextera Rapid Capture Exome Enrichment kit covers the entire exome area. The commercial next-generation sequencing panel appears to mean a gene panel targeting inherited retinal diseases (IRD). I think you need a more detailed description. For example, it is better to describe whether the panel and analysis pipeline includes IRD genes such as OAT, CHM and COL2A1(Stickler syndrome with posterior chorioretinal atrophy).
We rewrote in methods line 68: “The next generation sequencing panel, targeting inherited retinal diseases, including genes that cause chorioretinal dystrophies such as CHM, OAT, RPGR, CYP4V2, and more 200 genes were performed to establish a genetic diagnosis in patients 1, 3 and 4. CYP4V2 gene sequencing was performed by using Sanger sequencing in patient 2.”
line 128-129,
Genetic testing also ~.
It would be better to remove this awkward sentence that does not match the overall content. Instead, the authors can describe the presence or absence of family history, systemic diseases, and the results of family genotyping (some homozygous mutations on NGS may be due to exon or total gene deletion in one allele).
We have deleted the sentence and have added in line 127 “He denied family history”. We also found out that patient 3 had a history of consanguinity that we added in line 118 : “She referred parental consanguinity”
Line 181,
Even if the variant is predicted to be pathogenic or likely pathogenic in ACMG, this is just a prediction. Therefore, it cannot be determined that the new variant is the cause of the disease only by this prediction. In general, researchers report new phenotypes in patients with well-known pathogenic variants, or new pathogenic variants in patients with typical disease phenotypes. Authors should be cautious about describing a new variant found in a single patient as a cause of new phenotypes. For example, in Asians, even severe high myopia may be accompanied by diffuse chorioretinal atrophy and RD similar to case 4. It is necessary to describe the differential diagnosis, how the authors exclude other chorioretinal diseases, and what genes were excluded by genetic analysis. It would be better to add this point to the study limitations.
The missense variant c.1169G>T (p.Arg390Leu) found in patient 4 has never been reported. It is absent in all populations of the gnomAD database and was classified as disease-causing by nine pathogenicity predictors (DANN, GERP, FATHMM, LRT, MutationAssessor, Mutation Taster, PROVEANS, FATHMM-MKL, and SIFT). Furthermore, the Arg390 amino acid is highly conserved, and other missense variants have already been reported at the same codon (p.Arg390His and p.Arg390Cys). Thank you for your comments. We added a sentence in discussion line 209: “The fourth patient underwent a retinal dystrophy panel that included genes that cause other crystalline retinopathies such as retinitis punctata albescens (RLBP1), and fundus albipunctatus (RDH5), and genes that cause other chorioretinal diseases such as choroideremia (CHM), gyrate atrophy (OAT), and retinitis pigmentosa (RPGR). CTNS, ALDH3A2, COL4A5 that can cause cystinosis, Sjogren-Larson syndrome, and Alport syndrome were not tested. Only the novel variant c.1169G>T was identified. Additional cases with the same variant are missing to compare with our findings. Functional assessment of the novel missense variant may further elucidate its pathogenicity. Segregation analysis is not available in our cohort of patients.”

Reviewer 2 Report
The paper by Mariana da Palma et al. identified a new genotype and phenotype associated with Bietti crystalline dystrophy (BCD). The missense variant c.1169G>T (p.Arg390Leu) found in this paper has not been reported earlier. In addition, authors also report macular hole or retinal detachment in patients with BCD which progresses in different stages. The patient with the novel missense mutation presented with worst phenotype. Authors also, discuss the limitations of the study, where there is a need for substantial molecular analysis to elucidate the pathogenicity. The abstract is concise and accurately summarizes the essential information of the paper. Furthermore, the article is well constructed, the fundus images and mutational information provides the evidence for the conclusions stated. I have few suggestions for the authors though.
- Did authors perform electrocardiograms (ERGs) on the patients, if so, please provide the data in the manuscript.
- Please provide the values of the patient’s characteristics and OCT parameters in BCD patients in a tabular format.
Author Response
The paper by Mariana da Palma et al. identified a new genotype and phenotype associated with Bietti crystalline dystrophy (BCD). The missense variant c.1169G>T (p.Arg390Leu) found in this paper has not been reported earlier. In addition, authors also report macular hole or retinal detachment in patients with BCD which progresses in different stages. The patient with the novel missense mutation presented with worst phenotype. Authors also, discuss the limitations of the study, where there is a need for substantial molecular analysis to elucidate the pathogenicity. The abstract is concise and accurately summarizes the essential information of the paper. Furthermore, the article is well constructed, the fundus images and mutational information provides the evidence for the conclusions stated. I have few suggestions for the authors though.
- Did authors perform electrocardiograms (ERGs) on the patients, if so, please provide the data in the manuscript. Patient 1 has a report of ffERG on her chart review. The ffERG was normal. We added this information in the results, line 90: “The full field electroretinogram was normal at age 12”. We also added the following statement in methods, line 66: “Electroretinogram was performed in patient 1 in accordance to the International Society for Clinical Electrophysiology of Vision (ISCEV).
- Please provide the values of the patient’s characteristics and OCT parameters in BCD patients in a tabular format.
We added the following table in our manuscript.
|
Patient |
Age (years), gender |
c.DNA change in CYP4V2 |
Protein change |
BCVA OD OS |
Crystalline deposits |
OCT findings |
|
1 |
19, female |
c.802-8_810delinsGC homozygous |
p.? |
20/20 20/20 |
Retina |
Intraretinal hyperreflective crystals |
|
2 |
54, female |
c.518 T>G c.802-8_806del |
p.Leu173Trp p.? |
20/50 20/400 |
Retina |
Outer retinal atrophy, few intraretinal crystals, and tubulations |
|
3 |
69, female |
c.518T>G homozygous |
p.Leu174Trp |
20/150 20/150 |
Retina |
Not available |
|
4 |
59, male |
c.1169G>T homozygous |
p.Arg390Leu |
LP HM |
None |
Extensive atrophy and diffuse thinning. Central retinal detachment |

Reviewer 3 Report
A study by Mariana da Palma et al. (Manuscript ID: genes-1182987) presents four patients affected with molecularly confirmed Bietti crystalline dystrophy and suggests expanding phenotype of this condition.
Bietti crystalline dystrophy is a rare form of inherited retinal degenerations with higher prevalence in the Asian population. The authors presented ocular phenotype based on vision assessment and ophthalmic imaging. Further molecular diagnosis was established with commercial NGS panel. One of the presented patients (Patient 4) had novel homozygous variant c.1169G>T (p.Arg390Leu) and new clinical features, which have never been described before in the patients affected with Betti crystalline dystrophy.
This study has few main points:
- The study presents novel disease-causing variations and expanding the ocular phenotype for presented conditions (however it pathogenicity need to be proved in this case, at least e.g. in silico protein modelling).
- The quality if images is good.
- The authors highlight the importance of genetic testing in the conditions of overlapping phenotype, which is very important.
Weaknesses:
- Minor English language improvements should be addressed.
- Explanation of the pathogenicity of the novel variation is missing (see point 3. In the Broad comments). This is especially important because the patient with novel variants had also new clinical features, which were not previously described in the Bietti crystalline dystrophy.
Broad comments:
- Change “mutation” to “pathogenic variant”. This appears in the Discussion paragraph. Nomenclature should be written according to the latest recommendations of HGVS.
- Different retinal imaging modalities were used in this study. The authors should describe in the Methods paragraph which equipment was used.
- Do the authors have access to electrophysiological data and visual fields for studied patients? This would contribute significantly to the phenotype assessment, even if the data are described in the next.
- The pedigree and detailed family history is missing, especially important for Case 4, where described variant is novel and has atypical ocular changes. Since this patient is almost 60 y.o. probably segregation analysis is not possible in his parents. However if they are available for testing, this would improve the manuscript significantly. Would the authors consider functional studies or at least in silico protein modelling to explain the impact of novel mutation on the protein function and by this explaining pathogenicity?
- Figure 1, Patient 3 – is the OCT image available?
- Would the authors consider to make a one table with ocular findings for each presented case?
Specific comments:
- Abstract, row 13: “The ultra-rare Bietti crystalline dystrophy” – please change to “The rare form of retinal dystrophy, Bietti crystalline dystrophy”.
- Abstract, row 19-20: “The patient with the novel homozygous variant presented with the most severe phenotype with macular hole and retinal detachment in both eyes”- please change to e.g. “The patient with the novel homozygous variant had the most severe phenotype resulting in macular hole formation and retinal detachment in both eyes”.
- Abstract, row 22: add “had” before “the mildest phenotype…”
- Abstract, row 23: remove “older than the first patient”
- Introduction, row 42-43: change sentence to “The crystalline deposits seen early in the retina are the hallmark of BCD”.
- Introduction, row 45 – add other fleck retinopathies, such as dense deposit disease, retinal appearance in Alport syndrome, and support with appropriate references.
- Introduction, row 49-50: “This overlap emphasizes the importance of molecular testing to establish a precise diagnosis” – consider changing to e.g. ”The overlap of clinical phenotype, on both early and late stages of the disease, emphasizes the importance of molecular testing in establishing a precise diagnosis and has an impact on the disease management”.
- Material and Methods, row 58-60. Please change the sentence to: “The medical records were reviewed. The patients underwent detailed ophthalmic exams including best-corrected visual acuity (BCVA), slit-lamp exam and multimodal retinal imaging: fundus images (using equipment..), FAF (with equipment…)OCT (using equipment…). The commercial next generation sequencing panel ….was performed to establish a genetic diagnosis…”
- Discussion, row 139: “very interethnic admixture” – please remove this phrase or change.
- Discussion, row 140-141: remove this sentence “It is expected to find BCD in our population”.
- Discussion, row 141: change “ultra-rare” to “rare”.
- Discussion, row 151: change “compared with” to “compared to”.
- Figure 1, row 84: add ages of the patients to the images. Change to “Autofluorescence showed hypo-autofluorescent dots representing the areas of atrophy”. Row 86: change to “photography”.
- Figure 2 , row 133: add patient’s age to the images. Change to “Fundus photography” or “Fundus image”. Row 135: change to “retinal and choroidal atrophy”.
Author Response
Broad comments:
- Change “mutation” to “pathogenic variant”. This appears in the Discussion paragraph. Nomenclature should be written according to the latest recommendations of HGVS. We changed “mutation” to “pathogenic variant” in the discussion paragraph
- Different retinal imaging modalities were used in this study. The authors should describe in the Methods paragraph which equipment was used. We added this information, line 62: “The patients underwent detailed ophthalmic exams including best-corrected visual acuity (BCVA), slit-lamp exam and multimodal retinal imaging: fundus images (VISUCAM 500, Zeiss, Oberkochen, Germany), FAF (HRA2, Heidelberg Engineering), OCT (Spectralis, Heidelberg Engineering, Heidelberg, Germany). Electroretinogram was performed in patient 1 in accordance to the International Society for Clinical Electrophysiology of Vision (ISCEV).
- Do the authors have access to electrophysiological data and visual fields for studied patients? This would contribute significantly to the phenotype assessment, even if the data are described in the next. Unfortunately, we have ERG and microperimetry only for the first patient. The fourth patient could not fix to do microperimetry. The second and third patient had no functional testing. We added this information in the results, line 90: “The full field electroretinogram was normal at age 12”.
- The pedigree and detailed family history is missing, especially important for Case 4, where described variant is novel and has atypical ocular changes. Since this patient is almost 60 y.o. probably segregation analysis is not possible in his parents. However if they are available for testing, this would improve the manuscript significantly. Would the authors consider functional studies or at least in silico protein modelling to explain the impact of novel mutation on the protein function and by this explaining pathogenicity? We agree that segregation analysis is important. We added in discussion, line 217: “Segregation analysis is not available in our cohort of patients.” The novel missense variant c.1169G>T (p.Arg390Leu) found in patient 4 is absent in all populations of the gnomAD database and was classified as disease-causing by pathogenicity predictors. Furthermore, the Arg390 amino acid is highly conserved, and other missense variants have already been reported at the same codon (p.Arg390His and p.Arg390Cys). All this together reinforces the pathogenicity of p.Arg390Leu. We agree that functional studies are important to elucidate the pathogenicity of this novel variant. We added in discussion, line 216: “Functional assessment of the novel missense variant may further elucidate its pathogenicity”. Unfortunately, our team cannot perform these functional studies.
- Figure 1, Patient 3 – is the OCT image available? The quality of the image is really bad. We agree that the quality of images is not good. Unfortunately, patient 3 has no OCT images.
- Would the authors consider to make a one table with ocular findings for each presented case? We reviewed all data with the lab, we reviewed the genotype results and we corrected the patient 2 final results.
|
Patient |
Age (years), gender |
c.DNA change in CYP4V2 |
Protein change |
BCVA OD OS |
Crystalline deposits |
OCT findings |
|
1 |
19, female |
c.802-8_810delinsGC homozygous |
p.? |
20/20 20/20 |
Retina |
Intraretinal hyperreflective crystals |
|
2 |
54, female |
c.518 T>G c.802-8_806del |
p.Leu173Trp p.? |
20/50 20/400 |
Retina |
Outer retinal atrophy, few intraretinal crystals, and tubulations |
|
3 |
69, female |
c.518T>G homozygous |
p.Leu174Trp |
20/150 20/150 |
Retina |
Not available |
|
4 |
59, male |
c.1169G>T homozygous |
p.Arg390Leu |
LP HM |
None |
Extensive atrophy and diffuse thinning. Central retinal detachment |
Specific comments:
- Abstract, row 13: “The ultra-rare Bietti crystalline dystrophy” – please change to “The rare form of retinal dystrophy, Bietti crystalline dystrophy”. We changed the sentence in the manuscript as suggested.
- Abstract, row 19-20: “The patient with the novel homozygous variant presented with the most severe phenotype with macular hole and retinal detachment in both eyes”- please change to e.g. “The patient with the novel homozygous variant had the most severe phenotype resulting in macular hole formation and retinal detachment in both eyes”. We changed the sentence in the manuscript as suggested.
- Abstract, row 22: add “had” before “the mildest phenotype…” We added “had” as suggested.
- Abstract, row 23: remove “older than the first patient” We removed the sentence in the manuscript as suggested.
- Introduction, row 42-43: change sentence to “The crystalline deposits seen early in the retina are the hallmark of BCD”. We changed the sentence in the manuscript as suggested.
- Introduction, row 45 – add other fleck retinopathies, such as dense deposit disease, retinal appearance in Alport syndrome, and support with appropriate references. We added the sentence, line 46: “Perimacular yellow-white dots are also seen in Alport syndrome”. We also added in discussion, line 192: “Giant macular hole has been already described in other coalescing fleck retinopathy, Alport syndrome.”
- Introduction, row 49-50: “This overlap emphasizes the importance of molecular testing to establish a precise diagnosis” – consider changing to e.g. ”The overlap of clinical phenotype, on both early and late stages of the disease, emphasizes the importance of molecular testing in establishing a precise diagnosis and has an impact on the disease management”. We changed the sentence in the manuscript as suggested.
- Material and Methods, row 58-60. Please change the sentence to: “The medical records were reviewed. The patients underwent detailed ophthalmic exams including best-corrected visual acuity (BCVA), slit-lamp exam and multimodal retinal imaging: fundus images (using equipment..), FAF (with equipment…)OCT (using equipment…). The commercial next generation sequencing panel ….was performed to establish a genetic diagnosis. We rewrote the sentence, line 62: “The medical records were reviewed. The patients underwent detailed ophthalmic exams including best-corrected visual acuity (BCVA), slit-lamp exam and multimodal retinal imaging: fundus images (VISUCAM 500, Zeiss, Oberkochen, Germany), FAF (HRA2, Heidelberg Engineering), OCT (Spectralis, Heidelberg Engineering, Heidelberg, Germany). Electroretinogram was performed in patient 1 in accordance to the International Society for Clinical Electrophysiology of Vision (ISCEV). The next generation sequencing panel, targeting inherited retinal diseases, including genes that cause chorioretinal dystrophies such as CHM, OAT, RPGR, CYP4V2, and more 200 genes were performed to establish a genetic diagnosis in patients 1, 3 and 4. CYP4V2 gene sequencing was performed by using Sanger sequencing in patient 2.”
- Discussion, row 139: “very interethnic admixture” – please remove this phrase or change. We have removed this phrase in the manuscript as suggested.
- Discussion, row 140-141: remove this sentence “It is expected to find BCD in our population”. We removed this sentence in the manuscript as suggested.
- Discussion, row 141: change “ultra-rare” to “rare”. We changed the word in the manuscript as suggested.
- Discussion, row 151: change “compared with” to “compared to”. We have made this change.
- Figure 1, row 84: add ages of the patients to the images. Change to “Autofluorescence showed hypo-autofluorescent dots representing the areas of atrophy”. Row 86: change to “photography”. We have made these changes.
- Figure 2 , row 133: add patient’s age to the images. Change to “Fundus photography” or “Fundus image”. Row 135: change to “retinal and choroidal atrophy”. We have made these changes.

Round 2
Reviewer 1 Report
The manuscript is well revised.